# 'It was a joint plan we worked out together'. How the I-WOTCH programme enabled people with chronic non-malignant pain to taper their opioids: a process evaluation

Vivien P Nichols ⬤ ,[1] Charles Abraham,[2] Sam Eldabe,[3] Harbinder Kaur Sandhu ⬤ ,[1] Martin Underwood ⬤ ,[1] Kate Seers ⬤ ,[4] On behalf of the I-WOTCH team

¹Clinical Trials Unit, University of Warwick Warwick Medical School, Coventry, UK
²School of Psychology, Burwood Campus, Deakin University, Melbourne, New South Wales, Australia
³The James Cook University Hospital, Middlesbrough, UK
⁴Warwick Medical School, Warwick University, Coventry, UK

**Correspondence to**
Professor Kate Seers;
Kate.seers@warwick.ac.uk

## ABSTRACT

**Background** The Improving the Wellbeing of people with Opioid Treated CHronic pain (I-WOTCH) randomised controlled trial found that a group-based educational intervention to support people using strong opioids for chronic non-malignant pain helped a significant proportion of people to stop or decrease opioid use with no increase in pain-related disability. We report a linked process evaluation of the group-based intervention evaluated in comparison to a usual-care control group that received a self-help booklet and relaxation CD.

**Methods** We interviewed 18 intervention facilitators, and 20 intervention and 20 control participants who had chronic non-malignant pain and were recruited from general (family) practices in the UK. Quantitative data included change mechanism questions on the trial questionnaires which explored motivation, expectations and self-efficacy. Fidelity was assessed by listening to a sample of audio-recorded group sessions and nurse consultations. Quantitative and qualitative data were integrated using 'follow a thread' and a mixed-methods matrix.

**Findings** Four overarching themes emerged: (1) the right time to taper, (2) the backdrop of a life with chronic pain, (3) needing support and (4) the benefits of being in a group. Delivery fidelity was good, adherence (83%) and competence (79%) across a range of intervention groups. Staff delivering the intervention found three typical responses to the intervention: resistance, open to trying and feeling it was not the right time. The group experience was important to those in the intervention arm. It provided people with a forum in which to learn about the current thinking about opioid usage and its effects. It also gave them examples of how feasible or personally relevant coming off opioids might be.

**Conclusion** The process evaluation data showed that the I-WOTCH intervention was well delivered, well received and useful for most interviewees. Being 'the right time' to taper and having support throughout tapering, emerged as important factors within the context of living with chronic pain.

**Trial registration number** ISRCTN49470934.

## STRENGTHS AND LIMITATIONS OF THIS STUDY

⇒ The success of I-WOTCH was demonstrated from multiple perspectives providing an insight into the acceptability, replicability and transferability of the intervention.

⇒ Enablers of, and barriers to, the tapering process were explored, both of which are valuable for tailoring of this type of intervention to the individual.

⇒ This protocol-based process evaluation showed good fidelity of delivery, at least minimal compliance in 62%, and a very positive experience of the intervention, suggesting its suitability for service rollout.

⇒ Although general practitioners screened patients and prescribed for tapering, their perspectives were not a part of this process evaluation but may have provided useful information for future implementation.

⇒ As 95% of participants in the process evaluation were white, the reach of the study was thus limited and given interviews were 12 months after the intervention, recall bias may be a limitation.

## BACKGROUND

Embedded process evaluations of complex interventions in randomised controlled trials (RCTs) are critical to increase understanding of how and why interventions may or may not work, as well as how practical interventions are when implemented in service contexts. They increase confidence in trial results and provide guidance on their implementation.[1 2]

In the landmark Improving the Wellbeing of people with Opioid Treated CHronic pain (I-WOTCH) trial (n=608), we demonstrated that a self-management support intervention helps people using strong opioids become opioid-free at 1 year: 29% intervention versus 7% control, OR 6.55 (95% CI 3.42 to 12.55) with no increase in pain interference.

The I-WOTCH intervention included educational, behavioural and psychological components, combining group and one-to-one support. Papers reporting the intervention development, the trial protocol, the main results and process evaluation protocol have been published.[3–6] Here we report the I-WOTCH process evaluation, specifically.

► Experiences of the I-WOTCH intervention(s): including enablers of, and barriers to, change among participants.

► Implementation of the I-WOTCH group intervention: exploring the dose delivered and received, and the fidelity of delivery.

► Change mechanisms potentially underpinning intervention effects.

► Contextual issues: exploring how these may affect the outcome or running of the study and/or intervention.

## METHODS

We explored experiences of the participants and intervention delivery staff as well as key areas of process evaluation: context (contextual factors which may affect the implementation), fidelity (whether the intervention was delivered as conceived), dose delivered (the amount of the intervention delivered) and dose received (the amount of the intervention received by participants).[7] The group sessions took place once a week for 3 weeks. There was a face-to-face consultation between the second and third group sessions. After the third group session, there were two telephone conversations and then a final face-to-face consultation over a total period of 9–10 weeks. The process evaluation ran alongside the main study from May 2017 to March 2020.

The I-WOTCH trial recruited participants with chronic non-malignant pain from general (family) practices from midlands and northeast of England (table 1). Inclusion and exclusion criteria are listed in box 1. Once randomised all participants received a self-help booklet 'My Opioid Manager' and a relaxation CD. The intervention group also received a 3-day group intervention, run by a specially trained nurse/other HCPs and lay person with chronic pain who had tapered their own opioids, with two additional face-to-face nurse consultations and telephone support.

We used process evaluation theory to devise a logic model specific to this intervention and its theoretical underpinnings, informed by The Information, Motivation and Behaviour Skills Model[8 9] which highlights that knowledge, strong and stable motivation, and prerequisite skills are needed to initiate and maintain behaviour change (online supplemental material 1—logic model). We used a mixed-methods approach for the process evaluation (table 1).

### Interviews

Potential interviewees (participants and staff) were sent an information leaflet. Those interested gave informed consent to be interviewed. Intervention delivery staff were told of the interviews at their training and were approached on completing their final group. Semi-structured interviews were recorded using an encrypted device (OLYMPUS DS-7000 digital voice recorder), see online supplemental material 2 for indicative topic guide. Recordings were transcribed verbatim, anonymised, given numerical identifiers to maintain confidentiality and checked for accuracy by the research interviewer (VPN). The data were analysed using NVivo V.12 software to explore and organise the data. Warwick CTU's (clinical trials unit) lone worker policy, including a risk assessment, was followed for interview visits.

All data were collected and stored in digitally secure locations with restricted access in accordance with the Data Protection Act 2018c12 which is the UK's implementation of the General Data Protection Regulation.[10]

### Participant feedback forms

We gave feedback forms to the last 10 groups of participants after completing day 3 of the intervention. Questions asked about different aspects including how the group intervention was delivered and what they had found personally useful/not useful.

### Dose delivered and dose received

We collated trial data including date, location and attendance of the intervention groups to determine the dose delivered and received.

### Fidelity

Our fidelity assessment is detailed in our protocol paper and online supplemental material 3 shows our random sampling.[5] We listened to audio recordings of 11 prespecified, intervention group sessions across days 1, 2 and 3 (those which were educational and promoted discussion see online supplemental material 4) to assess adherence to the manual and competence of delivery. Other more practical sessions were checked for content presence but not rated in terms of delivery competence. We used bespoke assessment forms for group and individual sessions across a range of facilitators. Adherence items were scored as taking place: yes (2), partially (1) or no (0). Competence items were scored as: evident (2), partially evident (1) or not evident (0) (see online supplemental material 5). Scores were summated and averages calculated for adherence and competence of each session.

A random 10% of first and second one-to-one nurse/patient consultation recordings were also assessed throughout the study. If audio recordings were missing or inaudible, then the same sessions as those missing would be taken from the next group or one-to-one consultation in the same area (northeast or midlands). A second researcher (KS) double coded 10% of all the assessments.

### Potential change mechanisms

Four change mechanism questions about: patient motivations, expectations, self-efficacy and perceived credibility of the intervention were added to the main trial

**Table 1** I-WOTCH process evaluation: aims, key components, methods, analyses and data type

| Aims addressed | Key components | Methods | Analysis | Data type |
|---|---|---|---|---|
| Experiences of being in the I-WOTCH study | Participant interviews | Semistructured, face-to-face at home or convenient location, after they had completed the post 12-month questionnaire for the main study. Purposively sampled to attain a range across; age, opioid reduction experience, location and gender across the intervention and control arms. n=20 intervention and n=20 control group. | Framework analysis[24] | Qualitative |
| Experiences of delivering the I-WOTCH Intervention | Intervention delivery staff interviews | Semistructured, face-to-face at workplace or convenient location or telephone. After all intervention groups were completed. Pragmatic sample of nurse and lay facilitators—all invited. n=18 | Framework analysis[24] | Qualitative |
| Experiences of attending the group components of the intervention | Participant feedback forms | Paper form given to last 10 groups after day 3, asked to complete and send back in trial return prepaid envelope. Open and closed questions. n=31 | Thematic analysis[25] | Qualitative and quantitative |
| Intervention implementation | Dose delivered | Trial data describing how many group sessions were conducted in each location. | Descriptive statistics, charts, tables or figures using STATA V.17.0. The mean and SD were presented for continuous data, and the frequency and percentage for categorial data, summarised by treatment arm. | Quantitative |
| Intervention implementation | Dose received | Trial data of attendance and attrition. Uptake of the one-to-one consultations and follow-up telephone calls. (It was not possible to record telephone conversations). | Descriptive statistics, charts, tables or figures using STATA V.17.0. The mean and SD presented for continuous data, and the frequency and percentage for categorial data, summarised by treatment arm. | Quantitative |
| Intervention implementation | Fidelity | All intervention groups and one-to-one nurse consultations audio recorded. A sample of recordings listened to and assessed for adherence and competence. | Descriptive statistics—adherence and competency scores summed, and percentage score calculated. | Quantitative |
| Potential effects of change mechanisms | Change mechanism questions | Trial data collected from four questions about participant motivation, expectations, self-efficacy and perceived credibility of the intervention in the main trial questionnaire. | Descriptive statistics, charts, tables or figures using STATA V.17.0. The mean and SD presented/continuous data, and the frequency and percentage/categorial data, summarised by treatment arm. | Quantitative |
| Contextual issues | Context | Contextual factors were considered across all the data | Thematic analysis | Qualitative and quantitative |
| (1, 2, 3 and 4) | Synthesis of the data | Mixed-methods approach combining all the data to produce a model of overarching themes. | Use of a 'mixed-methods matrix' and 'following a thread' analysis strategy. O'Cathain et al[26] | Qualitative and quantitative |

questionnaire to ascertain if any of these factors had an effect on opioid reduction (online supplemental material 6).

### The process evaluation team

The process evaluation team included: KS, a co-applicant and work package lead who has a long track record in qualitative and mixed-methods research in complex interventions, CA, a co-applicant, highly experienced in behaviour change research and process evaluations and VPN, a research fellow with a background in physiotherapy, experienced in mixed-methods in rehabilitation

RCTs. The team collated data and analysed it separately from the main study. The initial report was given to the senior and chief investigators SE, HKS and MU prior to the results from the main trial being available.

### Patient and public involvement statement

Patient and public involvement input was not used directly with this process evaluation although one lay advisor, recruited via UNTRAP (Universities/User Teaching and Research Action Partnership) who helped with the development of I-WOTCH and its intervention, stayed on to become a valuable team member. Study findings will be

---

**Box 1    I-WOTCH participant inclusion and exclusion criteria**

**Inclusion criteria**
⇒ Provision of written informed consent.
⇒ Aged 18 years old or above.
⇒ Using opioids for chronic non-malignant pain.
⇒ Using strong opioids for at least 3 months.
⇒ Using strong opioids on most days in the preceding month.
⇒ Fluent in written and spoken English.
⇒ Able to attend group sessions.
⇒ Willing for GP to be informed of participation.

**Exclusion criteria**
⇒ Regular use of injected opioid drugs.
⇒ Chronic headache as the dominant painful disorder.
⇒ Serious mental health problems that preclude participation in a group intervention.
⇒ Previous entry or randomisation in the present trial.
⇒ Participation in a clinical trial of an investigational medicinal product in the last 90 days.
⇒ Pregnant at time of eligibility assessment, or actively trying to become pregnant.
⇒ People receiving strong opioid for the management of pain due to active malignant disease.

---

disseminated via newsletters and a lay summary put onto the study website as well as feedback to UNTRAP and other partnerships as outlined in the I-WOTCH protocol paper.[5]

## FINDINGS
### Experiences of the I-WOTCH interventions
We had multiple sources of qualitative data (figure 1).

### Participant interviews
The 40 interviewees (20 control/20 intervention) had a mean age of 65, range 59–72 years, 38/40 (95%) were white and 25 (63%) were women. Half (50%) had been on opioids for >5 years and 31 (78%) had pain for >5 years. See online supplemental materials 7 and 8 for full interviewee characteristics and

compared with the main study demographics, noting the interview samples were broadly consistent with the main study. Eighteen participants declined an interview mostly because of 'health issues' (eg, appointments or operations) or 'not a good time' (eg, work commitments, too busy).

Each interviewee has been given a numerical identifier, which follows each presented quotation. Table 2 provides evidence from the control interviews.

### Control interviews
The majority (14/20) found the self-help booklet informative, useful for further resources they could access. Three did not remember receiving the self-learning manual and two found some of it difficult to understand.

Most (n=10) listened to the CD once or twice and did not find it useful. Another four who used it more than once or twice said that ultimately it was not useful for them. However, five participants used it to good effect over the trial period, some on a regular basis and others 'as and when' they felt they needed it.

### *Enablers and barriers to tapering*
In the interview schedule, we asked all interviewees about enablers and barriers to tapering. We analysed the data from all 40 patient interviews using framework analysis for enablers and barriers to the tapering process including any attribution to the I-WOTCH study from the participants' perspectives.

As we are focussing on the I-WOTCH intervention for this paper, we illustrate the themes with quotations from the intervention arm only for clarity.

### Enablers
We identified four themes: (1) readiness to start tapering, (2) I-WOTCH as a trigger or motivator, (3) continuing to taper (once they had started) and (4) living without opioids. Subthemes are listed in table 3 with exemplar quotes from the intervention participants.

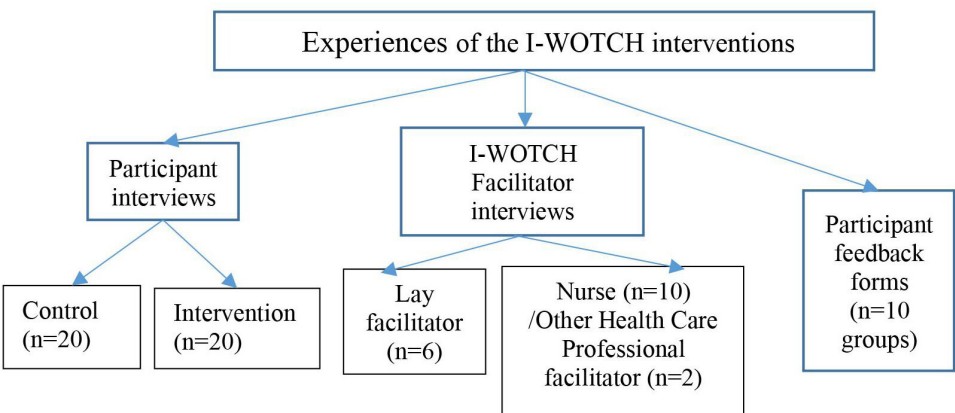

**Figure 1** Qualitative data sources. I-WOTCH, Improving the Wellbeing of people with Opioid Treated CHronic pain.

**Table 2** Control exemplar quotes

| Aspects of control intervention | Exemplar quotes |
|---|---|
| Some spoke about the information being a potential trigger to start the tapering process and that 'it made you think': Some found the information about risks and possible side effects interesting or shocking. | '…to make you think about what you're taking and why you're taking it'. 08. '…I was never told about the constipation problems that it would cause and obviously you read about the side-effects in the little leaflets and then it's put away!' 20 'sleep apnoea… that was a bit scary.' 23 |
| A few interviewees used the self-help booklet to start tapering. | 'I just followed everything it was doing…I just came down off the pills…I've done more exercise…' 15 |
| A few didn't find it as useful. | '…in a way it's assuming that you're overweight and you don't exercise!……it didn't tell me what I could do if I wasn't overweight…and I'm getting loads of exercise…… It doesn't really tell you what you can do as an alternative…' 28 |

### Barriers to tapering

Three barrier themes are from different phases of tapering: pre-tapering, during tapering and after stopping opioids. Subthemes are given in table 4A–C with exemplar quotes from the intervention.

### The role of the general practitioner (GP)

Most interviewees spoke about their GP in relation to their opioid use and tapering and the role of GP is presented in table 5. In the I-WOTCH study, intervention participants were given a tapering plan which was also sent to their GP and people were encouraged to make an appointment and discuss the plan. It was the GP who then actioned the plan if acceptable to both parties. The self-help booklet also suggested control participants could discuss tapering with their GPs. We have thus used quotations from intervention and control groups for this section on the role of the GP. There were five themes: (1) GP as information giver and sounding board, (2) trust in doctor, (3) teamwork, (4) GP reluctance and (5) access to GP. Table 5 gives exemplar quotes.

### Intervention interviews—participants' experiences of the intervention (n=20)

We present four main themes with subthemes where appropriate.
1. Acceptability of the intervention.
2. Being in a group.
   – Shared experience.
   – Social comparisons.
   – Support.
   – Being committed to something.
   – Group numbers.
   – Perceptions about group facilitators.
   – Challenges to establishing group cohesion.
3. Group sessions of interest.
   – Opioid information.
   – Distraction techniques.
   – Anger irritability and frustration.
   – Relaxation.
   – Mindfulness.
4. Nurse one-to-one consultations: participant perspectives.

### Acceptability of the intervention

Half said that the intervention was delivered in an appropriate format. When asked about suggestions to improve it, key responses were: provision of more support especially as tapering progressed, more group time, less didactic delivery and more local venues. All agreed they would recommend the I-WOTCH approach to others although some noted that participants needed to be open and receptive.

### Being in a group

Looking back on the group intervention, most interviewees found that being in a group was beneficial. They spoke about the groups as a shared experience where they were with people in similar circumstances which included the lay facilitator with whom they could share experiences. There were social comparisons within the group with participants noticing that others were like, or not like, themselves. Social support from the group was a key theme with evidence of people encouraging others especially when they were experiencing difficulties. Interviewees often talked about the commitment to the group and or the trial. Group numbers seemed important, less than three members limited discussion. Lay group facilitators were seen either as good examples of how people cope with chronic pain or, alternatively, that they were somehow not the same and may not have had the same long-term experience of opioids and pain. The former perception allowing greater identification with the lay facilitator. Not all groups ran smoothly especially if they included disruptive individuals (figure 2).

### Group session aspects of interest

There were five sessions which people talked about repeatedly across the data (see box 2) but on the whole people tended to talk globally about the intervention even when prompted to discuss individual sessions.

### Nurse one-to-one consultations: participant perspective

Most intervention participants talked about these sessions. A majority appreciated the tailoring of the

**Table 3** Enablers to tapering opioids

**(A) Theme 1: readiness to taper**

| Theme 1 subthemes | Exemplar quotes (with participant number)—intervention |
|---|---|
| Already made the decision or had started to taper | 'and then the invitation to take part in the study so I was reasonably invested in actually engaging with the study because it was something that I'd been thinking of myself… if you understand…' 29 |
| Didn't like the side effects | 'so it was a case of well I would try and see if I could reduce them or come off them and see if I wouldn't need as much other medication so it's the side-effect medication so I wouldn't be taking the anti-nausea tablets I wouldn't be having the… mmm… bowel medication…' 34 |
| Not wanting to be on opioids | '… it was when it was on the tele as well about being an opioid I thought I don't want to be addicted or to be totally dependent on them.'13 |
| Wondering if opioids were working anyway | 'Do you really need to be on this now? ……so I was reasonably invested in actually engaging with the study because it was something that I'd been thinking of myself' 29 |
| Support from GP or family to taper | '…I myself thought right I don't want to be on these anymore and the doctor agreed…' 25 |
| Positive past experience of tapering | '16–17 years ago just after I'd had my first pair of hip replacements before they started giving me problems I came straight off all the opioids I was taking then so I knew that I didn't really have a problem with… withdrawal from a previous experience…' 22 |
| The right time to fit with life events | 'I'm trying to think back to that part of the summer… how I was like… think I was relatively well which is why I started reducing it…' 34 |

**(B) Theme 2: I-WOTCH as a trigger or motivator to taper**

| Theme 2 subthemes | Exemplar quotes—intervention |
|---|---|
| I-WOTCH as trigger | '… but being part of the study has… mmm… enabled me not to be on them anymore whereas I don't know whether that would've been the case otherwise!' 29 |
| Information | '… and then when I was on the course I realised how damaging these things could be and I got all this information off the ladies and went through that …. well there's only one thing you've got to do… you've got to come off these damn things…' 30 |
| Support from group or nurse | '…I think because I had the support …. I just felt encouraged like that I could do it meself so I liked the support I suppose and the fact that they believed in me…' 13 |
| Tools to help | 'I was over the moon because I got more tips you know… … it was all helpful to me.' 10 |
| Tapering plan—given the means | '… it was a joint plan that we worked out together…' 09 |
| Open Mindset 'give it a go' | '… but I went in with an open mind and everything we did on the group the… the practice sessions and the things we got to do at home you know the colouring and everything else that we did on the study… mmm… and listening to the relaxation CD's and everything like that… I'm… I went in with an open mind on that and listened to it and took out what… what I could….' 09 |

**(C) Theme 3: continuing to taper**

| Theme 3 subthemes | Exemplar quotes—intervention |
|---|---|
| Self-efficacy | '… the Tramadol thing is still there I look at it at times and I sort of say 'maybe it's better that it's there instead of me throwing it away because that's training me' it's giving me enough courage to sort of say 'I'm straight I can actually see it and overcome it' and… err… I'm almost like it's like a mini training moulding myself on how to… to… to manage that…' 38 |
| Feeling better, decreased side effects | 'but it was really important to me to get rid of this… rid myself really of this drug… all I can say is this complete… having from the other side now I describe as 'un-reality'' 29 |
| Pain similar | '… but really I can say coming off of it the pain's not any worse than it was…' 31 |
| Seeing the reduction of opioids | '…so it just shows you… you just… you know if I can do without half of them I'm going to try and get down a bit further…' 21 |
| Flexibility (slow) of the tapering plan | 'it was done so… so slowly and minutely it was… I… I can't honestly say I had any side-effects!' 31 |
| Participant in control | 'I thought that was helpful you know to set your goals and know what you'd set' 31 |
| Understanding/weathering withdrawal effects | 'I was kicking I was twitching I was… I couldn't sit still I was… Aah… me hands were going and… I'm not joking I had to run round… I was running the table or walking fast round the table 'cause I couldn't keep still… I couldn't get any comfort whatsoever or any satisfaction and I was like 'Jeeze' and a couple of times I went… thought shall I put a patch on but I no I've got to persevere because if I put a patch on I know it will stop but I am back to square one so no I've got to persevere… I've got to persevere, and I did and I carried on and that's it… eventually it dispersed and… great so I've never touched the patches since… it's really great I'm chuffed…' 05 |
| Support from group or nurse | '…and that's what was good about the course with the support and the support network …… but it's that conversation among yourselves because you all know you're in the same boat you know what I mean and a lot of ya will have… would've had similar experiences and similar thoughts and whatever it be… err… it's… it's that which I think motivates ya…' 22 |

**(D) Theme 4: living without opioids**

| Living without opioids | '……because me body was getting itself put right if you understand where I'm at… …and in my mind I know I'm not abusing my body any more than what just for Paracetamols and everything I'm on now…' 10 |
|---|---|
| | '…there's negative points of coming off them but I feel the plus side is that these tablets… these opioids are not doing any harm to me body anymore…' 31 |
| | '… because of the course you know the few weeks that we were on that was a great help to me it just urged me to get off you know… wean myself off them which I did!' 25 |

GP, general practitioner.

**Table 4** Barriers to tapering

**(A) Theme 1: pre-tapering**

| Subthemes | Intervention—exemplar quotes |
|---|---|
| No intention: not receptive/'I'm not addicted'/ Not motivated | '… I think I'd be in a bad place if I couldn't take them… I certainly don't think I'm addicted to opioids' 16 |
| No choice, no alternatives | '… I really don't want to be on them that's my wish but… mmm… it's taught me that I cannot be without them which is unfortunate….' 25 |
| Not the right time: other health priorities /won't fit in with life (holidays)/awaiting procedures which may help or affect their pain for example, knee operation/operative pain | '…… I'd not even considered that at all until… until I have the… my hip replaced and then that knee's done that hip's done me eyes are done then we can see… start tapering off maybe having one… one a day instead of two a day.' 16 |
| Fear of: Increased pain/Decreased function/Not coping/Opioids being stopped/Having no future access to opioids | 'I remember thinking 'I'll be in so much pain and I'll be… I'll be even more angry with myself' so I was thinking, ' I really… I can't… I can't do this…' 21 |
| Resistance to group intervention messages: anger at GPs (Affecting GP/Patient relationship)/ Scepticism | '… but I'm quite annoyed that a GP who's prescribing more and more and more and should ought to… jolly well ought to know the research background so… and there was a lot of anger in the room that I was in about that…' 29<br>'but I was a little pessimistic… mmm… because of knowing how much pain I've been in… in the past and nothing else has worked…' 07 |

**(B) Theme 2: during tapering**

| Subthemes | Intervention—exemplar quotes |
|---|---|
| Increased pain | 'I think I was wrong to come off them believe it or not… and I really want to get off them because that make one very constipated and I just wanted to be free of them but I couldn't manage it… Saturday and Sunday were dreadful days I was in such pain!' 25 |
| Unpleasant withdrawal symptoms | '… but I'd say the symptoms were far worse when I reduced the second half …… … cramps in my legs and the… oh the hot sweats and cold sweats they went on for a long time that was horrible…' 01 |
| Tapering too fast | '… it's a shame that there wasn't another way or to have changed the patches in some way to have come down at a slower rate and perhaps then I may have had more success I think with coming down 25% because this particular patch that I'm on… the Butec one……it only comes in 5's…' 07 |
| GP unsupportive of the process or poor access | '…cos the doctor actually… err… didn't seem at all conversant with the idea…' 32 |
| Relatives worried not supportive of process | '… even my brother noticed he said 'you know it's been really rough for you coming off …' 07 |
| Missing trial support | 'I know that I couldn't have done it without going into a group to begin with to have that information if someone had just sat me down and said 'you're doing to reduce this by this and this' I would be screaming I think… no, no I can't you know yeah.' 21 |

**(C) Theme 3: after stopping opioids**

| Subtheme | Intervention—exemplar quotes |
|---|---|
| New or returned pain | '… I came completely off the opioids while I was doing the course… completely off them… erm… stopped them all… err… sadly I fell over in December… it caused me a load of problems I slipped a disc and whatever and hurt me left hip… err… so I went back on them to deal with the initial pain and if I'm completely honest with myself the pain's now bearable again but I've gone to that what I call my traditional crutch that I rely on…' 22 |

GP, general practitioner.

tapering plan and how they could personally apply the elements they had learnt from the intervention. Sessions were described as encouraging and supportive and almost all talked about having a good relationship with the nurse.

> … she was hopeful she said I had the determination to succeed in coming off the Tramadol… erm… so I felt positive with that… that I had like sort of the support with the words that I could actually do it … … I just felt encouraged like that I could do it meself so I liked the support I suppose and the fact that they believed in me…yeah! 13

### Participant feedback forms

Participants were given feedback forms at the end of day 3 in a subset of 10 intervention groups, providing 31 responses (delays in submitting an amendment about this for ethical approval meant the earlier groups were not able to be asked to complete this feedback form). Overall, responses indicate positive appreciation of most aspects of the course, but this is a small subsample, reported in detail in online supplemental material 9 and should be treated with caution.

### Intervention delivery staff interviews
#### Staff characteristics

We interviewed 18 intervention-delivery staff; nurse facilitators n=10, lay facilitators n=6 and other healthcare professional (HCP) facilitators n=2. Four (one nurse, three lay) did not respond. Identifier prefixes after each quotation are N for nurses, L for lay and H for HCPs.

We have presented the four themes from the staff interviews: (1) acceptability and training, (2) running the group sessions (venues, facilitators and group

**Table 5** Role of the GP

| Themes | Explanation | Exemplar quotes |
|---|---|---|
| GP as information giver and sounding board. | Prior discussion with GPs helped participants to prepare. Often this meant more interest or confidence in I-WOTCH. | 'Talking through (with the GP) made me realise I was ready' 4 control |
| Trust in doctor | Participants spoke of a history of trying everything else before opioids and often upward titration. They accepted this as they trusted their doctor. | 'I sort of trusted my doctor that he was doing the right thing by me…' 28 control |
| Teamwork | Some participants described a good working relationship with their doctor who supported and encouraged tapering. | '… I'd come down an awful lot and my doctor was absolutely gobsmacked… she calls me 'her star patient…'…she's a fantastic GP…. she wanted me to succeed as much as I did so it was teamwork but if you've got a good GP like that you're sailing aren't you but some are too busy!' 31 intervention |
| GP reluctance | There were some participants whose doctor was resistant to them tapering. | 'they didn't think I would get off them' (opioids) (because of their pain) 13 intervention<br>'(You)Can't come off them we've tried all sorts' 15 control |
| Access to GP | Some participants felt it was difficult to get to see their GP. | '… getting help really it's very difficult and getting an appointment up here is very difficult!' 20 control |

GP, general practitioner.

dynamics), (3) factors affecting readiness to change across group sessions and (4) one-to-one sessions.

*Acceptability and training*
Almost all were happy with the package as delivered. Quality assurance assessment on day 1 was appreciated as was the general support from the team 'I wanted that there was somebody there on the first day actually because it was quite good to find out how… whether I was actually doing it [correctly]…' L05

Staff interviewees regarded the training as adequate and the manual as comprehensive.

The manual 'was very good because it took you through every single section very, very clearly.' (N09)

'….it's quite thorough and really, really detailed.' L03

Some would have liked more training on; practical facilitation and/or subjects with which they were not familiar, for example, mindfulness or teaching posture. Most put in extra unpaid time to read around relevant subjects and to practice their delivery, and most were happy doing this.

*Running the sessions*
► Venues varied greatly, particularly in community settings. There were instances when even working to a lone worker risk-assessment protocol, facilitators were asked to lock up, being last in the building. On a few occasions, telephone back up with the I-WOTCH team was compromised by a lack of signal. IT problems were mitigated by facilitators having written scripts as backup.
► Facilitators mostly worked well together, with only occasional disagreements over differences in delivery or choice of sessions. Most would have preferred to work in the same pairings which

was not possible within the intervention delivery design. Nurses felt the days were quite 'full on' having little downtime because even in breaks and meal-breaks everybody still chatted. Lay facilitators who were having to deal with their own health issues commented that the travel and full days (09:00 to 15:00) left them tired. However, all the facilitators said that the lay facilitator role was key to the groups running well. '… do not remove the lay facilitators if you're going forward, they have to be there…' N02
► Group dynamics
There was sometimes initial scepticism and negativity about the programme which was challenging for the staff to deal with. Resistance sometimes changed around day 2.

… I think by sort of early afternoon is when people started you know listening a bit more thinking, 'Oh ok maybe there is an alternative?' L07

Groups fared differently—occasionally groups bonded or 'gelled' as early as day 1 but more often by the end of day 2 and definitely by day 3 where shared experiences and supportive bonds had been formed.

Now dealing with positive people looking and moving forward. L04

The most challenging groups involved a small minority of participants actively resistant to the information and techniques offered. Some showed challenging behaviour such as 'rubbishing' the content or interrupting. Some of these participants did not return after day 1.

'Shutters down' straight away. L05

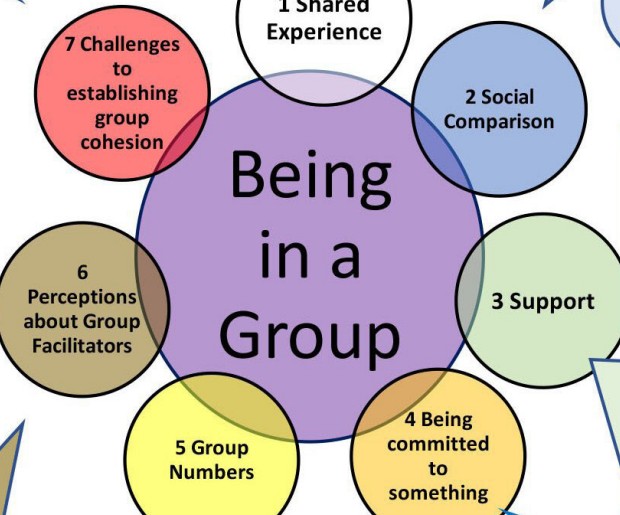

*... as a group we just accepted that this person was having issues and you accept them as part of the group ... 34*
*... I don't know but they were completely negative from the start to the finish... one guy particularly lasted until the coffee break and then never came back ...09*

*... you've got the same common factor amongst you... mmm... we've all got day to day pain and we are taking medication I think... because you know you've got that common bond...01*
*... I think hearing from the volunteer... the person who had come off and... ... she was brilliantly articulate about something that is really hard to articulate ... ...so I think that was so powerful to have somebody there who had experienced it.....29*

*...the thing that scared me the most was just when people were describing how opioids have affected them and you sort of say 'if I continue this is like me too' 38 Well I found it quite interesting to hear other people's tales... some people's tales were you know far worse than mine and yes it was good... 25*

*...from the way she was talking... it was a short term opioid problem she'd had... ...22*
*... the fact that you could actually see that sometimes he was struggling...... and yet he was coping...... managing his pain which is something that I knew that I would have to get out of the group as well... not just come off the Tramadol but also manage the pain after the Tramadol... 09*

*you can off load to somebody who understands so if you're going to a group with loads of people that have got a similar experience you feel like you are talking to people that understand ... 22*

*...there wasn't a lot of sort of interaction between I mean there was interaction between us but it wasn't as if it was a big group where you're bouncing ideas off each other all the time... ...30*

*... I felt as if I had the support...... I think I would've felt as if I'd failed both the study and the group and meself and I didn't want to I was determined I wanted to be off them! 13*

**Figure 2** Being in a group.

*Factors affecting readiness to change across group sessions—categories of change*

Intervention delivery staff described three presentations of change experience (across participants) and two turning points around the two main opioid information sessions. See boxes 3,4.

## Box 2 Specific aspects spoken about repeatedly.

**Aspects of interest**
*Opioid information:*
Information sessions gave most participants new insights into opioids and their effects. Participants also learnt about differences between dependence, tolerance and addiction and the effects of withdrawal and/or tapering and valued exploring these in the groups.
*Distraction techniques:*
Distraction techniques were felt to be the most useful technique, some recognised the technique in existing pastimes and hobbies.
*Anger irritability and frustration:*
This session was appreciated because participants felt they were not alone in having these feelings and that they had an opportunity to talk about them.
*Relaxation:*
Most liked the relaxation sessions but few used relaxation techniques regularly, as needed. Participants often confused relaxation with mindfulness.
*Mindfulness:*
Unless exposed to this previously, this was not well understood and often thought to be another form of relaxation. A few participants remained unsure about what it was.

### One-to-one sessions: nurse facilitator perspective
Nurse facilitators spoke of the tapering app being straightforward to use, although at times they were unable to get a mobile phone signal and occasionally the opioid the patient was taking was not included in the tapering app. Hard copies of the tapering plan were also written. On discussing the tapering plans suggested on the app algorithm, nurses and participants often wanted to taper more slowly for example, tramadol tablets decreasing by 50mgs but with no smaller dosages available, lack of confidence or having withdrawal effects. This was supported and nurses felt that it was important that the participant felt in control of the process.

> …a lot of the participants … mmm… didn't feel that they could taper at that speed and my sort of take on that is that it is a marathon not a sprint and you know if you can't taper at that speed then you know we'll go a bit further …' N10

Nurses described participants wanting to talk about a lot of issues which they hadn't necessarily brought to the group and that they needed that opportunity to talk.

> it gives them an opportunity to talk openly with me and perhaps mention things that they haven't been able to mention within the group. N10

Some nurses spoke about seeing a difference in participants at their first and second face to face.

> …people used phrases like they'd felt they'd 'come from behind a curtain.' And they didn't seem quite so dazed… N06

## Box 3 Three categories of change experience identified by intervention delivery staff

**Resistance to change**
Some did not want to engage, either that they were sceptical about the information, had no side effects or were fearful of coming off opioids. This was sometimes due to bad experiences or felt reliant on them or wanted something else as a substitute.
'… that was the bit they all dreaded… mmm… and they were super nervous and a lot of them I think weren't quite sure whether they wanted to take that step…' L05
'some people seemed to have no intention of looking at other ways to deal with their pain. Not Ready. L04. People 'start off pessimistic" L05

**Open to trying**
Some participants were ready to try—motivated themselves or by the group or that they felt the medication perhaps wasn't working anyway. There was concern after learning more about opioids especially its effects, tolerance and dependence. Tapering slowly was a reassuring message rather than 'all or nothing' enabling participants to 'just give it a go.'
'and sometimes they would talk themselves into it and talk themselves out of it… it was… cos it's a step into the unknown and we tell them this is what could possibly affect you …it did make it worse I… … cos some of the patients felt that they were already dealing with quite a bit you know to start with…… what made it really good was…… the other facilitator the nurse said you know 'it need not be a dramatic drop it can be a gentle tapering thing' and I think when they saw the tapering that settled them down at the end…' L05
'Became friends…Reassurance that they are not alone' L07

**Not the right time**
Staff spoke about participants wanting to delay tapering until after an event: for example, Holiday or surgery. They felt some were not able to manage it at that time but could reduce a little and try again later.
'… there was only the one man at first on the first class who hadn't been on opioids long… didn't feel dependent on them yet but he said he'd learnt and he knew what to look out for so yes he would be you know aiming to come off them but not quite yet…' L04

Nurse data suggested that confidence, motivation and the participant being in control were key factors to consider in the tapering process.

### Implementation of the I-WOTCH group intervention
This section explores how the trial was implemented and the uptake of the components offered (figure 3).

### Groups run (dose delivered)
The trial ran 35 groups—20 in the midlands, 15 in the northeast, of England. At randomisation, the mean group size was 9, SD 2.9, median group size 9 and IQR 5–11

### Uptake and attendance (dose received)
Minimal compliance is defined as attending at least day 1 and the first one face-to-face consultation. There were 190 (62%) with at least minimal compliance. Full compliance is defined as attending at least days 1, 2 and 3, the first one-to-one consultation and at least one phone call. There were 144 participants with full compliance (47%).

**Box 4    Turning points identified by intervention delivery staff**

1) On the morning of day 1 some were shocked and unaware of the issues and surprised to learn opioids aren't helpful long term.
*Bit of an eye opener N09*
'I think they were surprised to learn that opioids aren't helpful long term… … they would've been I think of the view that… err… I need to take this because if I don't take it I'll be in even more pain rather than this doesn't seem to be working cos I'm still in pain after 20 years!' N11
'(Angry with Doctors] and I think they were quite cross that they were getting it and being giving it and the length of time they'd been given… mmm that didn't go down too well in any of the groups!' L05
2) On the morning of day 2 when they started to think how they may start tapering and asked more questions. This was after the second session about opioids which covered withdrawal and John's tapering story which worked well.
'…my opinion was that they really needed their hands holding if they were going to make the jump…' H2
'… and then on day two pretty much you can tell who is more open to it ……most of the time it was the majority of the group you'd have the one or two that were still sceptical during day two still kind of like asking us questions…' L03

We randomised 305 people to the group intervention with 166 (54%) attending all three group days. The first one-to-one session attendance was 190 (62%) and 131 (43%) attended both the first and second session. One hundred and sixty-seven (55%) received one or more telephone calls. Thirteen (4%) did not return after day one (as mentioned in the staff interviews).

Non-attendance at the first and subsequent sessions was high at 90 (30%) (full trial attendance see online supplemental material 10).

*Participant reasons for non-attendance throughout the study*
Reasons for non-attendance (if given) were poor health, competing work commitments and family interests.

### Fidelity

Fidelity score averages for the group sessions adherence was 83% (very good 81%–90% as rated by Borrelli *et al*[11]), with a range of 25–100 and a median of 88 and overall competence 79% (good 71%–80%), range 0–100, median 86.[11] One-to-one session scores were high with an adherence average percentage score of 91%, range 61–100 and competence an average of 93%, range 50–100 (excellent 91%–100%[11]) (online supplemental materials 11 and 12). The 25% adherence score was rated when there was a technical problem and the facilitator did not use the back-up scripts provided, so did not cover that section of the session as expected. The lack of facilitation skills denoted by a score of 0 on competence does suggest facilitation is a skill shared by most but not by all.

### Potential change mechanisms

The control and intervention scores are comparable over the items of each question at baseline.

The two baseline only questions (2 and 3) (online supplemental material 6) were: Baseline expectation: I expect that, in 4 months' time, I will have reduced my opioid use and Baseline self-efficacy I am confident I could reduce my opioid use a lot over 4 months which showed a trend of low expectations of successful tapering within 4 months and low self-efficacy/confidence in tapering (online supplemental material 13).

The motivation to reduce opioids question showed a trend for more participants wanting to cut down or stop their opioids at baseline. There is a weak trend showing the self-management group to be more slightly more motivated to continue tapering at later timepoints. However, this should be viewed with caution due to the high proportion of missing data for this item.

Our statistician ran statistical analyses on the full data collected; baseline 4, 8 and 12 months to look for any impact of responses to the change mechanism questions on main trial outcomes (see online supplemental material

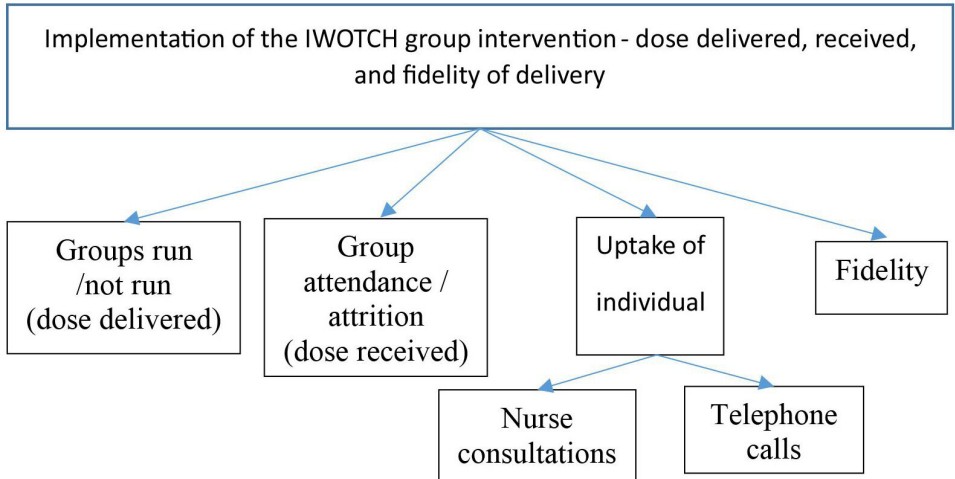

**Figure 3**    Structure of the implementation items. I-WOTCH, Improving the Wellbeing of people with Opioid Treated CHronic pain.

14). There were few significant correlations except for the study perceived credibility question. Correlations indicate that more participants attributed their tapering success to involvement in the intervention arm of I-WOTCH with the reverse trend in the control group (online supplemental material 15).

## Contextual issues

The main contextual issues were trying to live a life, often with multiple health problems and medications alongside managing multiple appointments, tests and operations. Day-to-day life is often a challenge dealing with their pain and the things they need to get done. Living with chronic pain can be challenging physically, mentally and emotionally. For some just the idea of starting to taper is 'a step too far' that they may be unwilling or unable to consider. This was seen in the participant interviews and the staff interviews especially with the three categories of change given previously in box 3 when people are: 'Resistant to change', 'Open to trying' or that it's 'Not the right time'.

## OVERARCHING THEMES FROM INTEGRATION OF QUALITATIVE AND QUANTITATIVE DATA

Analyses revealed threads across the qualitative and quantitative data which generated four main overarching themes: (1) the right time, (2) the backdrop of life with chronic pain, (3) needing support and (4) the group effect.

1. The right time. This theme encompasses evidence indicating whether someone is ready to change, including enablers or barriers to tapering and whether participants felt fully informed, motivated and confident to be able to taper.
2. Life with chronic pain is often complex with many having multiple health problems and medications. This means that participants were living life around their pain, pain relief, healthcare appointments and procedures as well as social and family commitments. Each of these could affect their decision about when or whether to taper.
3. Support is needed at all phases of tapering. This could come from their family, GP or from a tapering intervention such as I-WOTCH which agreed a tailored tapering programme. If family members or their GPs were ambivalent or against tapering, people were less keen to try it.
4. The group effect allowed for people taking opioid medication to come together to gain more information and skills in pain management as well as choosing whether to start on a tapering path as a joint group goal. They were given the opportunity to discuss and explore their fears and motivations about decreasing their medication.

The interaction between the overarching themes can be seen in our model. See figure 4. Participants make decisions about tapering based on whether it is the right time for them, their current pain experiences and healthcare and the support available to them. Helping participants to identify and discuss potential barriers while fostering knowledge, motivation, confidence and self-management skills means that tapering is more likely to take place.

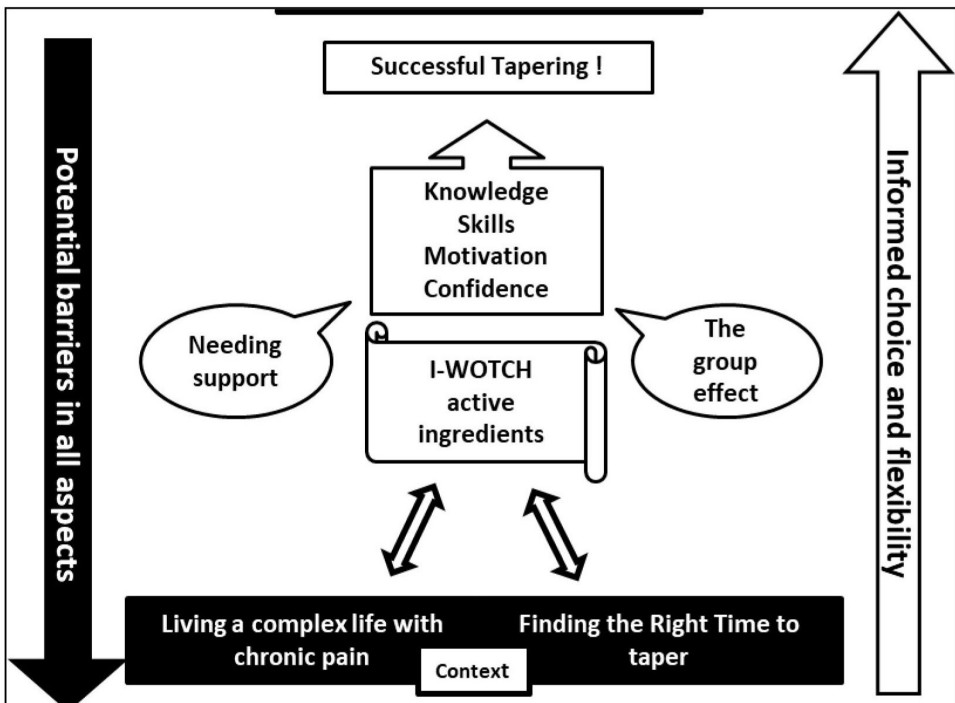

**Figure 4** Conceptual model of overarching themes. I-WOTCH, Improving the Wellbeing of people with Opioid Treated CHronic pain.

## DISCUSSION

The I-WOTCH main trial results showed that 62/225 (29%) of participants in the intervention group stopped taking opioids, compared with 15/208 (7%) in the usual-care group at 12 month follow-up without any difference in pain, or pain related disability, between the study arms. The process evaluation data indicate that the intervention was delivered as intended with good fidelity scores. There was evidence of participants attributing their opioid reduction to the I-WOTCH study. Participants and staff alike felt the package as whole was acceptable and deliverable. However, poor attendance was an issue with 90 participants randomised to the intervention not attending any group.

### The secret ingredient?

No one factor or 'secret ingredient' was identified but rather a combination of active ingredients as enablers of opioid tapering. The I-WOTCH intervention seemed to have a synergistic, summative effect from its different components which could be specific to the individual. The qualitative data indicate that the intervention served to inform and provide a forum where discussion in groups and in one-to-one sessions could help people decide whether to reduce their opioid use. The final decision was the participant's, and no judgements were made about whether or when they might start tapering.

The I-WOTCH intervention provided a space in which people could fill gaps in their knowledge of opioids. The intervention also enhanced motivations to taper, including intrinsic motivations that participants brought to the sessions. Information provided by the intervention about the lack of long-term effectiveness and side effects developed motivation. This was emphasised in staff interviews which highlighted how provision of information about opioid use consequences was, for some participants, a turning point in becoming motivated to change. Support from their GP, their family, the I-WOTCH one-to-one meetings, the tapering plan and group peers also enhanced motivation. Exploring pain management skills in a group over three sessions allowed them to try things out and swap ideas, so establishing the skill base needed for change.

The Information, Motivation and Behavioural Skills model[8 9] provided the theoretical foundation and guided our understanding of key components of change that would support action towards opioid tapering. The qualitative data confirm the utility of identifying sources of information, developing motivation at the right time and skill development, including use of the lay facilitator as a role model, as critical precursors of change. This process evaluation reveals how the I-WOTCH intervention provided all three of these critical supports. Barriers to change may be specific to the individual but supporting people to become informed and to develop motivation and skills optimises the chances of behaviour change. The main barriers were that it might not have been the right time for participants to consider tapering and

other things in life often took priority. Fears of returning to uncontrolled pain were common alongside multiple other potential barriers.

These findings correspond to a wider literature. A 2020 qualitative evidence synthesis using meta-ethnography (31 studies) looking at experiences of people taking opioids for chronic non-malignant pain by the current authors,[12] found five themes, four of which resonated with our findings. The fifth theme was around societal stigma, which did not emerge from the data in this English study but was common in North American studies, where law enforcement agencies can be involved.[13]

Our findings are also broadly supported by recent literature. Goesling et al who held focus groups with former opioid users about their experiences before, during and after opioid cessation.[14] Motivators to stop included concerns about a lack of opioid efficacy, addiction and quality of life impact. Quinlan et al noted that 60 patients undertaking a pre tapering survey found there were more barriers given than motivators.[15] The main areas of concern were around quality of life, pain and withdrawal which the authors suggest need to be addressed for successful tapering to take place. Henry et al conducted focus groups and interviews with 21 adults with chronic pain at different stages of opioid tapering.[16] They explored the complex contextual backdrop of life with chronic pain describing changes or fluctuations in many areas including pain, social relationships, health status, emotional state and the perceived need for opioids at any given time. They recognise the substantial effort involved for people undertaking the tapering journey, the effect on their day to day lives and the strategies used. They suggest early anticipatory guidance about tapering and that tapering should be patient centred and responsive to the patients' needs.

We identified GPs as integral to tapering support. Three 2020–2022 papers from healthcare providers perspective[17–19] throw light on the complex challenges involved in tapering opioids, especially the importance of maintaining a good patient/provider relationship. Hamilton et al also cite patient motivation to be a central factor for successful tapering.[19] GPs are seen as authoritative experts in relation to health behaviour and patients often welcome behaviour change advice from their GPs, especially when such advice may have a positive effect on long-term condition management.[20]

There is a long history of successful use of small groups to develop support, share common health problems, build trust, share ideas, enhance information, change attitudes and develop motivation and skills to promote health behaviour change.[21 22] This evaluation shows that the I-WOTCH intervention successfully applied these group processes in supporting individual change. Participants often had a strong sense of shared experience around opioid use and the challenges of use reduction. They felt as if they were all in the same boat. This is a key foundation for successful group interventions.[23] There is also evidence of participants experiencing social support

and social learning both key to individual change. Facilitation of interaction in small groups is critical to optimising change and interviews revealed that the lay facilitator role was very important to participants and nursing facilitators alike. Facilitators did not underestimate the challenges some participants might face and appreciated the lay facilitator's personal experience with which participants could identify and then learn from. The challenges of group facilitation observed also emphasise the importance of careful skill-based training for group facilitators.[22] At least minimum compliance was achieved with 62% of participants, and full compliance with 47%. All three group sessions were attended by 54%. We note that even with this level of compliance, clinically important differences were found. Even though adherence was less than we had hoped we have got a very clear positive result on one of the trial primary outcomes. It seems likely shared decision making between patients and GPs could be important to increasing compliance, although this would need to be evaluated. For future delivery, this study demonstrated that group sessions were an important part of the intervention, The one intervention that had the least positive feedback was mindfulness. However, we are reluctant to suggest this could be removed as not all facilitators felt comfortable explaining this element of the programme.

## CONCLUSION

Our conceptual model shows the active ingredients involved in an ideal pathway to successful tapering. This may look straightforward at first glance; however, the reality is that there are multiple contextual issues and barriers which influence someone's ability to taper. The I-WOTCH intervention gave practical strategies, information, support, room for discussion and a tailored tapering plan to help participants navigate this difficult journey.

**Acknowledgements** We would like to thank the participants and intervention delivery staff who were interviewed for their valuable input into the process evaluation. Also the members of the I-WOTCH team. Alleyne Sharisse, Balasubramanian Shyam, Betteley Lauren, Booth Katie, Carnes Dawn, Furlan Andrea D, Haywood Kirstie, Iglesias Urrutia Cynthia Paola, Lall Ranjit, Manca Andrea, Mistry Dipesh, Noyes Jennifer, Rahman Anisur, Shaw Jane, Tang Nicole KY, Taylor Stephanie, Tysall Colin, Underwood Martin, Withers Emma J.

**Collaborators** The I-WOTCH team: Alleyne Sharisse, Balasubramanian Shyam, Betteley Lauren, Booth Katie, Carnes Dawn, Furlan Andrea D, Haywood Kirstie, Iglesias Urrutia Cynthia Paola, Lall Ranjit, Manca Andrea, Mistry Dipesh, Noyes Jennifer, Rahman Anisur, Shaw Jane, Tang Nicole KY, Taylor Stephanie, Tysall Colin, Underwood Martin, Withers Emma J.

**Contributors** KS, MU, SE and HKS conceived the original design. VPN, KS and CA developed the study design and plan for data collection and undertook analysis. All authors have provided critical revisions to the manuscript and approved the final manuscript. KS is responsible for the overall content as the guarantor.

**Funding** This project is funded by the National Institute for Health Research (NIHR), Heath Technology Assessment (HTA) (project number 14/224/04). The views and opinions expressed therein are those of the authors and do not necessarily reflect those of the HTA, NIHR, NHS or the Department of Health.

**Competing interests** MU is chief investigator or co-investigator on multiple previous and current research grants from the UK National Institute for Health Research, and Arthritis Research UK and is a co-investigator on grants funded by the Australian NHMRC and Norwegian MRC. He was an NIHR Senior Investigator until March 2021. He is a director and shareholder of Clinvivo Ltd that provides electronic data collection for health services research. He receives some salary support from University Hospitals Coventry and Warwickshire He is a co-investigator on two current and one completed NIHR funded studies that are, of have had, additional support from Stryker Ltd. Until March 2020 he was an editor of the NIHR journal series, and a member of the NIHR Journal Editors Group, for which he received a fee. KS is chief or co-investigator on many previous and current grants from the UK National Institute for Health Research. She also sat on a NIHR funding board (HS&DR) until 2018. SE is a clinician and Chief investigator on several past and ongoing trials, his department has received research support from the NIHR as well as Medtronic, Boston Scientific and Saluda Medical. He is a paid consultant to Medtronic, Saluda Medical and Mainstay Medical.

**Patient and public involvement** Patients and/or the public were not involved in the design, or conduct, or reporting, or dissemination plans of this research.

**Patient consent for publication** Not applicable.

**Ethics approval** This study involves human participants and was approved by Yorkshire and the Humber-South Yorkshire Research Ethics Committee on 13 September 2016 (16/YH/0325). But this paper reports an embedded process evaluation. Participants gave informed consent to participate in the study before taking part.

**Provenance and peer review** Not commissioned; externally peer reviewed.

**Data availability statement** Data are available upon reasonable request.

**ORCID iDs**
Vivien P Nichols http://orcid.org/0000-0002-3372-1395
Harbinder Kaur Sandhu http://orcid.org/0000-0003-1522-8078
Martin Underwood http://orcid.org/0000-0002-0309-1708
Kate Seers http://orcid.org/0000-0001-7921-552X

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
