## [Reviewer comments · BMJ Open]

ARTICLE DETAILS

TITLE (PROVISIONAL)	"It was a joint plan we worked out together." How the I-WOTCH programme enabled people with chronic non-malignant pain to taper their opioids: a process evaluation
AUTHORS	Nichols, Vivien; Abraham, Charles; Eldabe, Sam; Sandhu, Harbinder; Underwood, Martin; Seers, Kate

VERSION 1 – REVIEW

REVIEWER	Glare, Paul University of Sydney Sydney Medical School
REVIEW RETURNED	08-Jun-2023

GENERAL COMMENTS	This is a very high-quality piece of research, very well written and I concur with their Conclusion. I learnt a lot from reading it and reviewing it. I have some expertise in opioid tapering research including the challenges faced by people with chronic pain who are now being asked/forced to taper long term opioid therapy that they believe has helped them previously. But I have never reviewed a process evaluation (PE) study before and I found it very challenging to review this paper, given its length, the multidimensional nature of PE, the mixed methods used, and the sheer amount of data presented. The task was simplified once I obtained the protocol paper and could check off what was done, and why. Table 1 is important for readers to understand how the study worked, serving the same function as the study schema figure for a clinical trial, so I think a much simpler version is needed here. I would move the current Table to the supplementary data (becoming piece number 16!) and replace it with a new Table 1 with subheadings for the different foci of PE (e.g. implementation data (dose, fidelity); experiences of participants and staff; putative mechanisms for achieving change; impact of contextual factors), putting the current Aims column first, followed by the Key Components, a column for the n's for each item (i.e. 40 for participant interviews) third, and the Data Type last. Strengths and limitations summary: it lists 3 Strengths and 2 Limitations (and the 2nd, the spectrum bias of the participants is a limitation of the clinical trial, not the PE). Limitations of the process are not mentioned in the Discussion and should be discussed there. In addition to the other limitation (non-involvement of GP), I think there is at least one other: they didn't address 'reach' of the intervention. Aside from the unavoidable fact that the participants were restricted to people who consented to be in the study, an evaluation of 'reach' would have focused on the fact that 95% participants were white. I did note also on p.23 line 50 they gave the mean for group size with the IQR, not the standard deviation. This should be corrected.
---

REVIEWER	Poole, Helen Liverpool John Moores University, School of Psychology
REVIEW RETURNED	14-Jun-2023

GENERAL COMMENTS	This paper reports a process evaluation of a complex intervention: I-WOTCH. It's great to see papers of this type which can continue to inform intervention development and help identify potential mechanisms of effect in complex interventions. The I-WOTCH intervention was designed to reduce opioid use in patients with chronic non-malignant pain and the trial protocol and outcome data have been reported in separate papers. The rationale and methods used are sound and represent a comprehensive for this type of evaluation. The authors collected data from a wide range of sources and provide a large quantity of supplementary material. I appreciate that the amount of data has made it difficult to write a succinct account of the evaluation and there are a number of pointers for additional information to be found in the other I-WOTCH papers throughout. Whilst useful, this is slightly irritating to the reader who has to search the referenced paper for the information. From my perspective it would be preferable to include such details in the extensive supplementary material. It's a good paper that describes the process evaluation well. Some specific comments for consideration below. P4 The retrospective nature of the interviews and recall bias should be considered as a limitation Reconsider wording of bullet point 3 in light of levels of compliance with the intervention. If take up is just over half in terms of compliance, is an intervention suitable for service roll out in its current format? P6 Make clear what time frame the intervention was run over - 1 day/week for 3 weeks? Include study start/end dates P7 The first row of the table includes reference to a post 12 month questionnaire – remove as not interviews In column 4 – analysis – is it necessary to include (sd) after standard deviation when it is not used as an abbreviation after the first mention? P8 Add brief explanation/detail here or in supplementary material 3 on random sampling – do the 1,2,3 relate to day numbers? P10 Participant interviews – could add statement about whether they consider the interview sample were broadly consistent with main study or not alongside reference to supplementary material. It would also be interesting to know what the compliance levels were like for those interviewed from the intervention group. State when patient interviews carried out – after their final group or after completion of intervention? Did all those in the control arm report using the self-help booklet? The statement in the bottom row of table 2 – it would be more informative to state number who reported using CD to good effect rather than 'some' Similarly, how many are the 'few who used it more than a small number of times'? P11
---

	Enablers and barriers to tapering: reword to make it clear that you interviewed 40 patients and analysed all, but here you present data from 20 in the intervention group only. P12-14 P12-14 & P20-22 I have some concerns about the themes and exemplar quotes in the Table 3abcd. I appreciate there is limited space for description of the themes and it's possible to misinterpret the labels and thus the overlap between subthemes. For example, in 3a Theme 1 sub-themes the much of the same quote is used to illustrate two of the sub-themes [already made the decision or had started to taper & wondering if opioids were working anyway]. Alongside, there are multiple quotes from the same participants [e.g. 29, 9]. With interview data from 20 intervention participants, providing quotes from across the dataset would give voice to more of the participants and increase confidence in the depth and extent of the analysis. This also applies to the findings presented from the delivery staff interviews [e.g. in box 3, 5] and in the one to one sessions, and I would encourage the authors to review this as above and include quotes from a wider range of participants. P23 State how many/what percentage achieved full compliance [n161] The narrative on fidelity notes a range for adherence to protocol of 25-100% and 0-100% for competence. The supplementary material shows sessions 2&3 on day 1 rather low. Could the authors speculate as to the reasons for the zero competence or low scores given other sessions show good fidelity overall? P26 Discussion: The authors note the process evaluation indicates the intervention was delivered as intended with good fidelity scores and they provide evidence in support of that as well as acknowledging how many did not attend any sessions. I'd like to see a bit more discussion around what the findings might mean for I-WOTCH or other similar interventions moving forwards. Some consideration/discussion of compliance levels is definitely warranted as just over half attended all three sessions. Such levels of non-compliance would be unpalatable to commissioners or in clinical practice. What are the implications of this for the wider roll out of an intervention like I-WOTCH? As no 'secret ingredient' was identified what recommendations can be made for adapting any future delivery ? some suggestions are made for key components like group delivery, but what elements might not be needed, e.g. would mindfulness still be included? Supplementary material 7 shows a range of Morphine Equivalent doses interviewees were on at baseline. There are some high doses and others relatively low/minimal, including zero. Are these consistent with the wider sample? Do the authors consider baseline dose impacted on patients experiences of the intervention at all? Can this be discussed further? P20-22 I have some concerns about the themes and exemplar quotes in the Table 3abcd. I appreciate there is limited space for description of the themes and it's possible to misinterpret the
--	---

	labels and thus the overlap between subthemes. For example, in 3a Theme 1 sub-themes the much of the same quote is used to illustrate two of the sub-themes [already made the decision or had started to taper & wondering if opioids were working anyway]. Alongside, there are multiple quotes from the same participants [e.g. 29, 9]. With interview data from 20 intervention participants, providing quotes from across the dataset would give voice to more of the participants and increase confidence in the depth and extent of the analysis. This also applies to the findings presented from the delivery staff interviews [e.g. in box 3, 5] and in the one to one sessions, and I would encourage the authors to review this as above and include quotes from a wider range of participants. P23 State how many/what percentage achieved full compliance [n161] The narrative on fidelity notes a range for adherence to protocol of 25-100% and 0-100% for competence. The supplementary material shows sessions 2&3 on day 1 rather low. Could the authors speculate as to the reasons for the zero competence or low scores given other sessions show good fidelity overall? P26 Discussion: The authors note the process evaluation indicates the intervention was delivered as intended with good fidelity scores and they provide evidence in support of that as well as acknowledging how many did not attend any sessions. I'd like to see a bit more discussion around what the findings might mean for I-WOTCH or other similar interventions moving forwards. As no 'secret ingredient' was identified what recommendations can be made for adapting any future delivery ? some suggestions are made for key components like group delivery, but what elements might not be needed, e.g. would mindfulness still be included? Supplementary material 7 shows a range of Morphine Equivalent doses interviewees were on at baseline. There are some high doses and others relatively low/minimal, including zero. Are these consistent with the wider sample? Do the authors consider baseline dose impacted on patients experiences of the intervention at all? Can this be discussed further? Some consideration/discussion of compliance levels is definitely warranted as just over half attended all three sessions. Such levels of non-compliance would be unpalatable to commissioners or in clinical practice. What are the implications of this for the wider roll out of an intervention like I-WOTCH?
--	---

VERSION 1 – AUTHOR RESPONSE

Reviewer 1		
1) This is a very high-quality piece of research, very well written and I concur with their Conclusion. I learnt a lot from reading it and reviewing it.	Thank-you, we appreciated your positive comments.	
2) Table 1 is important for readers to understand how the study worked, serving the same function as the study schema figure for a clinical trial, so I think a much simpler version is needed here. I would move the current Table to	Thank-you for this helpful comment. We have revised Table 1 and restructured the columns as you suggest. This has made the different foci of the PE much clearer, highlighted in bold. We	Table 1 (revised) on p6 track changes

the supplementary data (becoming piece number 16!) and replace it with a new Table 1 with subheadings for the different foci of PE (e.g. implementation data (dose, fidelity); experiences of participants and staff; putative mechanisms for achieving change; impact of contextual factors), putting the current Aims column first, followed by the Key Components, a column for the n's for each item (i.e. 40 for participant interviews) third, and the Data Type last.	have included the numbers in each group. We think these changes have made it much easier to understand and would like it to stay in the main text to help readers understand how the study worked.	
2) Limitations of the process are not mentioned in the Discussion and should be discussed there. Limitations: non-involvement of GP; 'reach' of the intervention - 95% participants were white.	We have included the reach of the intervention as a limitation and added the limitations of not interviewing GPs to the discussion.	An extra limitation has been added on p4 of track changes
3) p.23 line 50 they gave the mean for group size with the IQR, not the standard deviation.	Thank-you, the standard deviation has been added,	Added on p23
Reviewer 2		
4) It's a good paper that describes the process evaluation well.	Thank-you for your positive comments about the paper.	
5) p4 The retrospective nature of the interviews and recall bias should be considered as a limitation.	These have been added as a limitation	Recall bias has been added as a limitation (p4) in the strengths and weaknesses section
6) p4 Reconsider wording of bullet point 3 in strengths and limitations) in light of levels of compliance with the intervention. If take up is just over half in terms of compliance, is an intervention suitable for service roll out in its current format?	This reviewer raises an interesting point and we have discussed this carefully in the team. We do not think this would be unacceptable to commissioners or in clinical practice. That people may not attend a full course of a complex intervention of this nature is part of clinical life. For the group sessions it has little impact on service delivery as the session will run irrespective of whether an individual	

	person attends. Importantly even though adherence was less than we had hoped we have got a very clear positive result on one of the trial primary outcomes. Our view is that commissioners and clinicians will be much more focussed on this than the non-adherence. In practice one might expect better adherence outside the clinical trial scenario as we are offering an intervention of proven effectiveness rather than a speculative experimental procedure	
7) p6. Make clear what time frame the intervention was run over - 1 day/week for 3 weeks? Include study start/end dates	We have clarified this: "The group sessions took place once a week for three weeks. There was a face-to-face consultation between the second and third group sessions. After the third group session, there were two telephone conversations and then a final face to face consultation over a total period of 9-10 weeks. The process evaluation ran alongside the main study from May 2017 to March 2020.	Added to methods p6
8) p7. The first row of the table includes reference to a post 12 month questionnaire – remove as not interviews	Thank-you. We had not been clear enough – to clarify, interviews were "Semi-structured face to face at home or convenient location after they had completed the RCT's 12 month questionnaire, We have reworded this in Table 1.	Table 1 on p6
9) p7 In column 4 – analysis – is it necessary to include (sd) after standard deviation when it is not used as an abbreviation after the first mention?	Thank-you – we have removed sd as you suggest.	
10) p8. Add brief explanation/detail here or in supplementary material 3 on random sampling – do the 1,2,3 relate to day numbers?	Yes, we haven't been clear enough. The numbers relate to day 1, day 2 and day 3 groups. We have added the explanation below to the supplementary material 3.	See Supplementary material 3

	All of the intervention groups were given a chronological number. To identify which groups would be checked for fidelity, for each block of ten groups (representing early, mid and later stages of the study) the trial statistician randomly allocated a day 1, day 2 and day 3 session.	
11) p10 i) Participant interviews – could add statement about whether they consider the interview sample were broadly consistent with main study or not alongside reference to supplementary material. ii) State when patient interviews carried out – after their final group or after completion of intervention? iii) Did all those in the control arm report using the self-help booklet? iv) The statement in the bottom row of table 2 – it would be more informative to state number who reported using CD to good effect rather than ‘some’ Similarly, how many are the ‘few who used it more	i) We have added “The interview sample were broadly consistent with the main study, see supplementary materials 8. ii) Patient interviews were carried out after the 12 month follow up. This is now addressed in the revised text for Table 1 (see comment 8). Above Table 2 we have added: “The majority (14/20) found the self-help book informative, useful for further resources they could access. Three participants said that they didn’t remember receiving the self-learning manual and two found it difficult to understand. We initially had the numbers in, then took them out as we were reporting qualitative findings where counts are less useful. However, given this reviewer’s comments, we have reinstated the numbers. We have added this detail to table 2: “Most (10) listened to the CD once or twice and did not find it useful. Another four who used it	Added on p11 See table 1 p6 See p11

than a small number of times'?	more than once or twice said that ultimately it wasn't useful for them. However, five participants used it to good effect over the trial period, some on a regular basis and others 'as and when' they felt they needed it."	See Table 2 p12
12) p11 Enablers and barriers to tapering: reword to make it clear that you interviewed 40 patients and analysed all, but here you present data from 20 in the intervention group only.	We have reordered our text so it is clearer that we are presenting data from the 20 in the intervention group	See "enablers and barriers to tapering on p12
13) P12-14 & P20-22 i) With interview data from 20 intervention participants, providing quotes from across the dataset would give voice to more of the participants and increase confidence in the depth and extent of the analysis. This also applies to the findings presented from the delivery staff interviews [e.g. in box 3, 5] and in the one to one sessions, and I would encourage the authors to review this as above and include quotes from a wider range of participants.	Thank-you for highlighting this important point. We have reviewed the supporting quotations used and note 17/20 participants' quotations were used in the findings which we feel provides a diverse range of views. We agree we needed a wider range of quotations for the delivery staff interviews and these have now been included in Box 3 and in the main text.	Box 3, on p21 and in the text p19-22
14) P23 State how many/what percentage achieved full compliance The narrative on fidelity notes a range for adherence to protocol of 25-100% and 0-100% for competence. The supplementary material shows sessions 2&3 on day 1 rather low. Could the authors speculate as to the reasons for the zero competence or low scores given other sessions show good fidelity	We have added full compliance % to the text, The 25% adherence score was rated when there was a technical problem and the facilitator did not use the back-up scripts provided, so did not cover that section of the session as expected. The lack of facilitation skills denoted by a score of 0 on competence does suggest	Under uptake and attendance heading p23 Added p24

overall?	facilitation is a skill shared by most but not by all.	
15) Discussion: i) I'd like to see a bit more discussion around what the findings might mean for I-WOTCH or other similar interventions moving forwards. Some consideration/discussion of compliance levels is definitely warranted as just over half attended all three sessions. Such levels of non-compliance would be unpalatable to commissioners or in clinical practice. What are the implications of this for the wider roll out of an intervention like I-WOTCH? ii) what recommendations can be made for adapting any future delivery? some suggestions are made for key components like group delivery, but what elements might not be needed, e.g. would mindfulness still be included?	We have added to the discussion: i) At least minimum compliance was achieved with 62% of participants, and full compliance with 47%. All three group sessions were attended by 54%. We note that even with this level of compliance, clinically important differences were found. Even though adherence was less than we had hoped we have got a very clear positive result on one of the trial primary outcomes. It seems likely shared decision making between patients and GPs could be important to increasing compliance, although this would need to be evaluated. ii) Adaptions for future delivery This study demonstrated that group sessions were an important part of the intervention, The one intervention that had the least positive feedback was mindfulness. However, we are reluctant to suggest this could be removed as we also discovered that not all facilitators felt comfortable explaining this element of the programme. iii) Morphine equivalent doses. are broadly consistent with wider sample. We think your observation is an interesting one, but we are not able to determine whether the baseline dose impacted on experiences of the intervention as this study was not designed to assess this in a robust manner.	Added to discussion on p29 Added p29-30.

iii) Supplementary material 7 shows a range of Morphine Equivalent doses interviewees were on at baseline. There are some high doses and others relatively low/minimal, including zero. Are these consistent with the wider sample? Do the authors consider baseline dose impacted on patients experiences of the intervention at all? Can this be discussed further?		
--	--	--